# Highly porous metal-organic framework liquids and glasses via a solvent-assisted linker exchange strategy of ZIF-8

Wen-Long Xue [1], Pascal Kolodzeiski[1], Hanna Aucharova[2], Suresh Vasa[2], Athanasios Koutsianos[1], Roman Pallach [1], Jianbo Song [1], Louis Frentzel-Beyme [1], Rasmus Linser [2] & Sebastian Henke [1]✉

By combining the porosity of crystalline metal-organic frameworks (MOFs) with the unique processability of the liquid state, melt-quenched MOF glasses offer exciting opportunities for molecular separation. However, progress in this field is limited by two factors. Firstly, only very few MOFs melt at elevated temperatures and transform into stable glasses upon cooling the corresponding MOF liquid. Secondly, the MOF glasses obtained thus far feature only very small porosities and very small pore sizes. Here, we demonstrate solvent-assisted linker exchange (SALE) as a versatile method to prepare highly porous melt-quenched MOF glasses from the canonical ZIF-8. Two additional organic linkers are incorporated into the non-meltable ZIF-8, yielding high-entropy, linker-exchanged ZIF-8 derivatives undergoing crystal-to-liquid-to-glass phase transitions by thermal treatment. The ZIF-8 glasses demonstrate specific pore volumes of about 0.2 $cm^3 g^{-1}$, adsorb large amounts of technologically relevant $C_3$ and $C_4$ hydrocarbons, and feature high kinetic sorption selectivities for the separation of propylene from propane.

Metal-organic frameworks (MOFs) are network structures composed of organic linkers interconnecting inorganic building units (metal ions or (oxo)-clusters) via coordination bonds[1–3]. At present, nearly 100,000 unique MOFs have been reported, mainly in the form of small single crystals and polycrystalline powders[4]. Although MOFs in crystalline form have been used for various applications, their lack of processability and grain boundary effects limit their utility for the fabrication of homogeneous membranes for gas separation and efficient solid-electrolytes. Zeolitic imidazolate frameworks (ZIFs)[5] are an important subset of MOFs and major formers of the fourth generation of MOFs[6–8], namely liquid MOFs[9] and their corresponding glasses[10,11], offering processable bulk shapes which still retain the advantageous chemical functionality of crystalline MOFs. The advent of fourth-generation MOFs is proposing an elegant solution for the processing of these materials in bulk shapes of specific geometries, as required for

most practical applications, by moulding them in their (supercooled) liquid state[12,13].

However, only very few ZIFs have been observed to melt at elevated temperatures and to form glasses by melt-quenching. This is because the materials' decomposition temperature ($T_d$) is typically lower than their melting temperature ($T_m$). In most cases, the imidazolate-based organic linkers decompose before the metal-linker coordination bonds break and reorganize – the prerequisite for ZIF melting[9–11,14]. So far, only ZIFs with the relatively dense **cag, zni** and **gis** network topologies have been reported to melt, whereas ZIFs exhibiting more porous topologies (i.e. **sod** or **lta**) do not melt on their own but can be flux melted in the liquid of a meltable ZIF[14–16]. The most critical issue, however, is that the gas-accessible porosity of the known melt-quenched ZIF glasses is rather low and ranges between 14% and 16% of the bulk glass volume[17]. Moreover, the pores of the ZIF glasses

[1]Anorganische Chemie, Fakultät für Chemie und Chemische Biologie, Technische Universität Dortmund, Otto-Hahn Straße 6, 44227 Dortmund, Germany.
[2]Physikalische Chemie, Fakultät für Chemie und Chemische Biologie, Technische Universität Dortmund, Otto-Hahn-Straße 4a, 44227 Dortmund, Germany.
✉e-mail: sebastian.henke@tu-dortmund.de

are very narrow, so that N₂ molecules (kinetic diameter 3.6 Å) do not enter the intrinsic pore network of the glasses at 77 K, whereas the diffusion of the larger *n*-butane (kinetic diameter 4.3 Å) is kinetically hindered at 273 K, resulting in very low gas uptakes and strongly hysteretic sorption behaviour[11]. This greatly limits their application in gas adsorption and separation, guest molecule loading and transport, so the development of MOF glasses, especially melt-quenched MOF glasses with high porosity, remains an important task.

ZIF-8[18] (Zn(mim)₂), also known as MAF-4[19], is a high-porosity MOF (about 50% of the crystal volume is void) consisting of Zn²⁺ and 2-methylimidazolate (mim⁻) connected to a framework with large spherical cavities of about 1 nm diameter and **sod** (sodalite) topology. ZIF-8 is the single most extensively studied ZIF material, both from a

fundamental[20–22] and an applied[23–26] perspective. Nevertheless, ZIF-8 cannot be directly formed into a glass by melt-quenching as the free-energy barrier for melting is too high due to its high porosity[27]. Thus, crystalline ZIF-8 decomposes at about 540 °C before reaching the liquid state[10] It has been demonstrated, however, that infiltration of an ionic liquid (IL) into the pores of ZIF-8 yields a meltable and glass-forming IL@ZIF-8 composite[28]. Unfortunately, the composite partially decomposes during melting while the porosity of the derived IL@ZIF-8 glass is even smaller than the porosity of the prototypical ZIF-62 glass.

Here, we incorporate two additional organic linkers, im⁻ (imidazolate) and bim⁻ (benzimidazolate), in varying ratios into ZIF-8 by solvent-assisted linker exchange (SALE). The two additional linkers are supposed to play distinct roles in enabling the melting of ZIF-8 crystals.

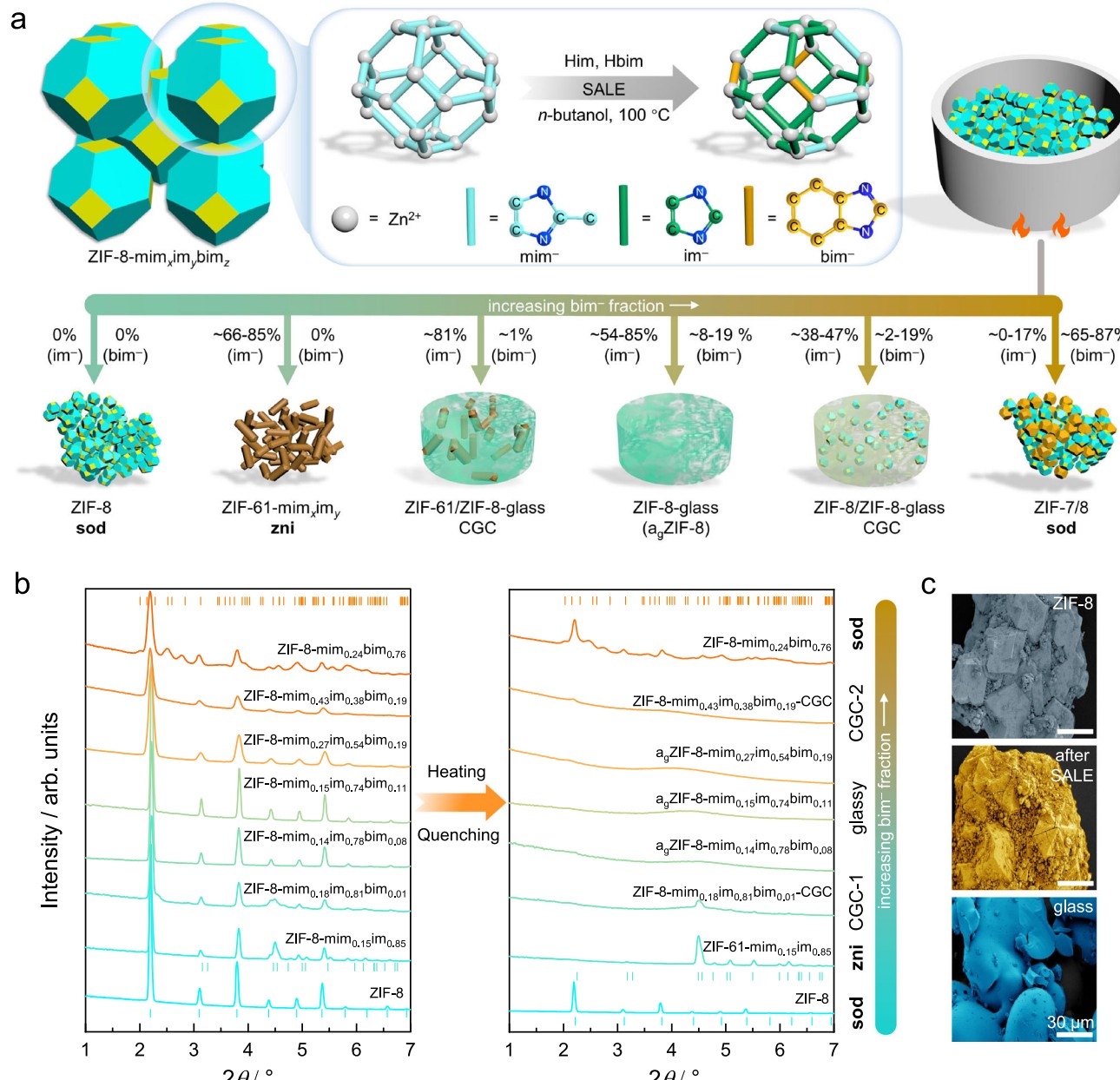

**Fig. 1 | Representation of the SALE process to derive various linker-exchanged ZIF-8 derivatives and their crystallographic and microscopic analysis.**
**a** Schematic of the SALE process leading to the incorporation of im⁻ and bim⁻ linkers in ZIF-8 and the thermal phase behaviour of the derived linker-exchanged ZIF-8 derivatives. **b** Synchrotron radiation XRPD patterns of selected ZIF-8 derivatives before (left) and after (right) heating to 414 °C (λ = 0.45920 Å). Tick marks represent the characteristic Bragg reflection positions of ZIF-7 (orange tick marks, CCDC code RIPNOV01), ZIF-61 (upper cyan tick marks, CCDC code GITTAF) and ZIF-8 (lower cyan tick marks, CCDC code FAWCEN). **c** False colour SEM images of crystalline ZIF-8, crystalline ZIF-8-mim₀.₁₅im₀.₇₄bim₀.₁₁ (after SALE) and glassy a₉ZIF-8-mim₀.₁₅im₀.₇₄bim₀.₁₁ (after melt-quenching, a₉ = amorphous glass), all scale bars are 30 μm.

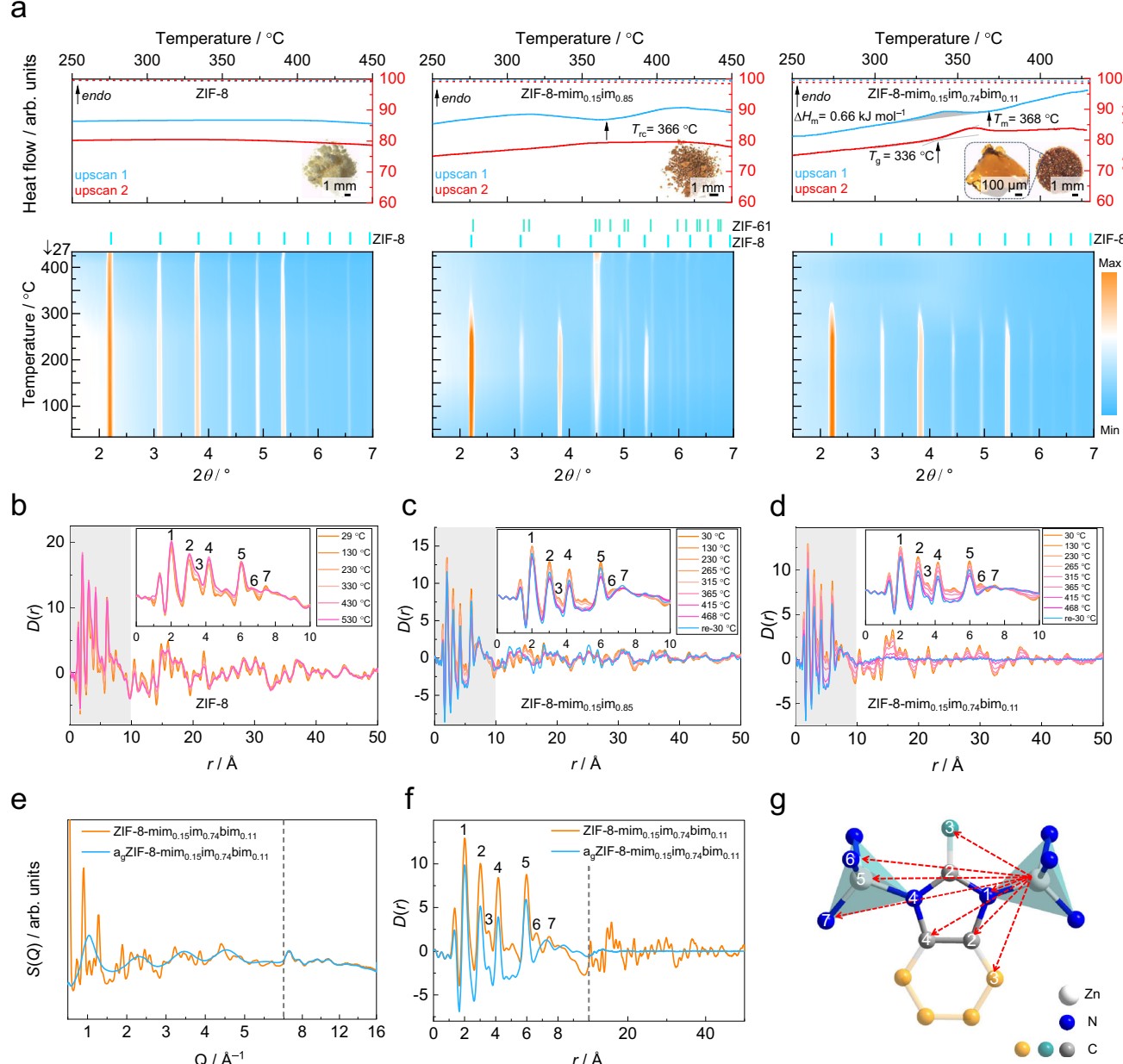

**Fig. 2 | Thermal analyses, in situ and ex situ synchrotron X-ray diffraction and scattering data. a** TGA (dotted lines, right vertical scale) and DSC (solid lines, left vertical scale) traces of ZIF-8 (left), ZIF-8-$mim_{0.15}im_{0.85}$ (middle) and $a_g$ZIF-8-$mim_{0.15}im_{0.74}bim_{0.11}$ (right, the grey shaded region highlights the integrated area for the determination of $\Delta H_m$). The samples were cycled from 50 °C to 450 °C or 430 °C with a heating/cooling rate of ±10 °C $min^{-1}$ twice. The insets show micrographs of ZIF-8, recrystallized ZIF-61-$mim_{0.15}im_{0.85}$, glassy $a_g$ZIF-8-$mim_{0.15}im_{0.74}bim_{0.11}$ derived by cooling the materials from 450 °C or 430 °C to room temperature. Contour maps of the VT-XRPD patterns are shown below the TGA/DSC data. Each map is generated from 11 XRPD patterns recorded at temperatures between 27 °C and 414 °C, containing 10 patterns recorded during heating and one additional pattern recorded after returning to room temperature. **b–d** VT-PDF data of ZIF-8 (**b**), ZIF-8-$mim_{0.15}im_{0.85}$ (**c**) and ZIF-8-$mim_{0.15}im_{0.74}bim_{0.11}$ (**d**) derived from VT X-ray total scattering experiments. The insets highlight changes in the local structure ($r < 10$ Å, grey-shaded region). **e** Room temperature X-ray total scattering data in the form $S(Q)$ of ZIF-8-$mim_{0.15}im_{0.74}bim_{0.11}$ and $a_g$ZIF-8-$mim_{0.15}im_{0.74}bim_{0.11}$. The latter was prepared by melt-quenching from 430 °C in the DSC apparatus. **f** PDFs in the form $D(r)$ obtained from the $S(Q)$ data shown in (**e**). **g** Visualization of the relevant short-range atomic distances present in the ZIFs and their assignment to the peaks in the PDFs. Since $mim^-$, $im^-$ and $bim^-$ are statistically distributed throughout the materials, the carbon atoms shown in cyan and orange are only partially occupied and thus have only a minor contribution to the PDFs.

The $im^-$ linker is known to exhibit weaker binding to $Zn^{2+}$ compared to the original $mim^-$ linker[27]. Conversely, the bulky $bim^-$ linker is essential for inhibiting the crystallization into a dense ZIF phase with the **zni** topology[11]. Depending on the precise chemical composition (i.e. the molar ratio of the three different linkers), some of the derived linker-exchanged ZIF-8 derivatives undergo crystal-to-liquid-to-glass phase transitions by thermal treatment. By analyzing altogether 50 linker-exchanged ZIF-8 samples with varying compositions, a triangular phase diagram is generated, depicting regions of (i) congruent melting, (ii) incongruent melting and (iii) decomposition before melting in the available phase space. The porosity of a representative congruent melting and glass-forming ZIF-8 derivative is quantified for the crystal and glass forms by $CO_2$ and $N_2$ gas sorption measurements. The ZIF-8 glass exhibits a substantially higher specific pore volume than the prototypical ZIF-62 glass. Moreover, gas physisorption experiments establish that the ZIF-8 glass adsorbs rather large amounts of

technologically relevant $C_3$ and $C_4$ hydrocarbons and features the ability to kinetically separate propylene from propane. This demonstrates an unanticipated potential of linker exchange for the generation of high-porosity MOF glasses for a broad range of technical applications. Importantly, the present work also reveals that MOF melting is not limited to crystalline precursors having rather dense network topologies (e.g., **cag** or **zni**), but can be extended to more porous MOF phases through targeted linker engineering and linker mixing.

## Results

### Preparation and characterization of linker-exchanged ZIF-8 derivatives

ZIF-8 (Zn(mim)$_2$) was synthesized by an established solvothermal procedure[21]. Subsequently, the SALE method was used to exchange a fraction of the mim$^-$ linkers for im$^-$ and bim$^-$ (Fig. 1a and Methods). By running the SALE reactions for different times (between 1 and 7 days) and with varying molar ratios of the linkers in $n$-butanol, 49 unique linker-exchanged ZIF-8 samples with the general composition Zn(mim)$_{2x}$(im)$_{2y}$(bim)$_{2z}$ ($x + y + z = 1$), here denoted ZIF-8-mim$_x$im$_y$bim$_z$ (Supplementary Table 1 and Supplementary Figs. 1, 2) were prepared.

Fourier transform (FT) IR spectra recorded on the activated ZIF-8 derivatives (heated at 100 °C under vacuum for 12 h) prove that neither $n$-butanol nor protonated linkers (i.e. Hmim, Him and Hbim) remain in the pores of the linker-exchanged ZIF-8 derivatives after activation (Supplementary Fig. 60). The molar ratio of the three linkers in the ZIF-8-mim$_x$im$_y$bim$_z$ derivatives was determined by solution $^1$H nuclear magnetic resonance (NMR) spectroscopy of acid-digested samples (Supplementary Figs. 4–15, Supplementary Table 1). Noteworthy, the mim$^-$ linkers in all the SALE products were intentionally not completely exchanged in any of the samples, so that a wide variety of sample compositions was generated by the described approach (Supplementary Fig. 2).

X-ray powder diffraction (XRPD) was used to investigate the phase identity of the ZIF-8-mim$_x$im$_y$bim$_z$ derivatives (Fig. 1b, Supplementary Figs. 17–28). Based on the diffraction data, the following conclusions can be drawn:

(i)   The **sod** structure of ZIF-8 is maintained after the SALE process, even though at some compositions a second minority phase is formed in parallel.

(ii)  If only im$^-$ is incorporated (i.e. if Hbim is not present in the exchange solution and $z = 0$) a small amount of a denser ZIF phase with **zni** topology is formed. The formed phase is related to ZIF-61, a dense ZIF with the nominal chemical composition Zn(mim)(im)[5].

(iii) With increasing amounts of bim$^-$ incorporation the fraction of the ZIF-61-like minority phase formed during the SALE reaction is decreasing.

(iv)  If larger amounts of bim$^-$ are incorporated ($z > 0.65$) a phase mixture of the cubic **sod** phase (typical for ZIF-8) and the heavily distorted rhombohedral or triclinic **sod** phase (typical for ZIF-7, Zn(bim)$_2$)[18,29] is present.

These main conclusions are supported by structure-less profile fitting (Pawley method[30]) of XRPD patterns using reference data from the literature (Supplementary Figs. 30–35, Supplementary Table 2). The profile fits further reveal that the unit cell volume of the cubic ZIF-8-mim$_x$im$_y$bim$_z$ phases is between 0.7% and 2.5% smaller than the unit cell volume of conventional ZIF-8, indicating that the lack of the methyl group in position 2 of the im$^-$ and bim$^-$ linkers allows for a slightly denser packing of the building units in the **sod** structure. It is worth noting that the SALE reaction mixtures must remain undisturbed throughout the linker exchange process. Notably, when the reaction mixture is stirred for 15 min per day during a 3-day SALE procedure, the original ZIF-8 crystals transition to the denser ZIF-62 phase, with only 5% of the linkers being mim$^-$ (yielding a composition of

Zn(mim)$_{0.10}$(im)$_{1.60}$(bim)$_{0.30}$; Supplementary Figs. 29 and 35). This finding underscores the critical role of slow linker exchange under static conditions for preserving the ZIF-8 structural integrity.

### Thermal behaviour

The thermal behaviour of the materials was comprehensively analyzed by differential scanning calorimetry (DSC), thermogravimetric analysis (TGA), variable temperature (VT)-XRPD and VT X-ray total scattering. Pair distribution functions (PDFs) in the form $D(r)$, providing information about the characteristic atom-atom correlations in the materials, were obtained from the corrected X-ray total scattering functions $S(Q)$ via Fourier transformation[31,32]. Similar to pristine ZIF-8, all linker-exchanged ZIF-8 derivatives have very high thermal stability with minimal weight-loss up to 500 °C and decomposition temperatures ($T_d$) above 500 °C as proven by TGA under $N_2$ atmosphere (Supplementary Figs. 63a–74a). Cyclic DSC measurements (two upscans, one downscan) were performed from room temperature to 430 °C or 450 °C to screen for potential melting (first upscans) and glass transition signals (second upscans) (Supplementary Figs. 63b–74b).

As expected, pristine ZIF-8 does not show any phase change up to at least 530 °C, which was confirmed by the retention of the Bragg peaks in VT-XRPD, the absence of thermal events in two consecutive DSC upscans (Fig. 2a left) and the persistence of long-range order in the VT-PDFs (Fig. 2b). However, ZIF-8 loses its high thermal stability after most of the mim$^-$ linkers are exchanged by im$^-$. Starting from around 310 °C, ZIF-8-mim$_{0.15}$im$_{0.85}$ undergoes an exothermic solid-state transition to a phase called ZIF-61-mim$_{0.15}$im$_{0.85}$ exhibiting the dense **zni** topology (Fig. 2a middle, Fig. 2c). The thermally triggered reconstructive phase transition of ZIF-8-mim$_{0.15}$im$_{0.85}$ from the **sod** to the **zni** phase indicates that the inclusion of im$^-$ facilitates metal-linker bond breaking[27]; the essential element of MOF melting. It is known that the thermally triggered reconstruction of **zni** phases can be prevented and direct melting facilitated by including larger linkers in the frameworks, i.e. linkers that are too large to be incorporated in the **zni** topology[11]. As expected, several ZIF-8-mim$_x$im$_y$bim$_z$ derivatives (represented by ZIF-8-mim$_{0.15}$im$_{0.74}$bim$_{0.11}$ in Fig. 2) possess well-defined melting points ($T_m$) in the range from 360 °C to 399 °C in the first DSC upscan (Supplementary Figs. 64–66, 74 and Supplementary Table 4), corroborated by the complete loss of Bragg scattering above 310 °C in VT-XRPD experiments (Fig. 2a right and Supplementary Figs. 41–44) and the loss of long-range order in the VT-PDFs (Fig. 2d and Supplementary Figs. 53, 54). The congruently melting ZIF-8-mim$_x$im$_y$bim$_z$ derivatives lie in a range of compositions featuring large fractions of im$^-$ ($0.54 \leq y \leq 0.85$) and small fractions of bim$^-$ ($0.08 \leq z \leq 0.19$). Importantly, crystallization is not detected upon cooling the liquid ZIF-8-mim$_x$im$_y$bim$_z$ derivatives to room temperature (Fig. 2e–g), while a second DSC upscan displays clear glass transition events with glass transition temperature ($T_g$) values in the range from 334 °C to 361 °C, establishing the formation of a$_g$ZIF-8-mim$_x$im$_y$bim$_z$ (a$_g$ = amorphous glass).

The influence of the linker ratios on the thermal behaviour of the ZIF-8 derivatives is also evident from the morphological changes during heat treatment (Supplementary Methods 7). Pristine ZIF-8 and ZIF-8-mim$_{0.15}$im$_{0.85}$ remain as polycrystalline powders after heating to 450 °C. In contrast, the particles of ZIF-8-mim$_{0.15}$im$_{0.74}$bim$_{0.11}$ fuse into a monolithic glassy structure when heated to 430 °C (Fig. 2a inserts). Scanning electron microscopy (SEM) imaging (Fig. 1c) reveals that the microcrystals of pristine ZIF-8 develop some cracks on the crystal facets after the formation of ZIF-8-mim$_{0.15}$im$_{0.74}$bim$_{0.11}$ via SALE. The cracks originate from chemical stress upon linker substitution via the solid-liquid reaction[21]. Conversely, a$_g$ZIF-8-mim$_{0.15}$im$_{0.74}$bim$_{0.11}$ derived from melt-quenching exhibits the morphological characteristics of a typical ZIF glass, showing particle coalescence to a fused structure which evidently experienced significant flow while in the liquid state.

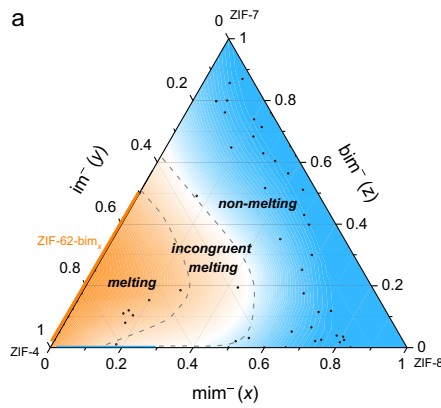
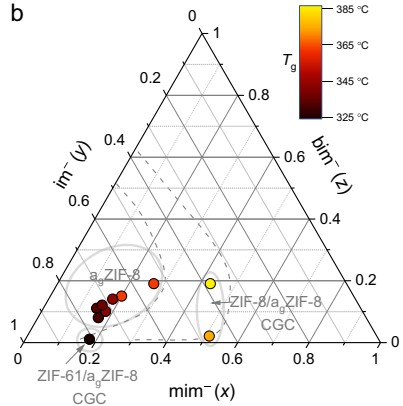

**Fig. 3 | Ternary phase diagrams of ZIF-8-mim$_x$im$_y$bim$_z$ displaying the melting and glass transition properties. a** Ternary phase diagram based on thermal analysis and XRPD data of 50 derivates of ZIF-8-mim$_x$im$_y$bim$_z$ along with literature data of ZIF-4[10] and ZIF-62-bim$_x$ (orange line)[11,50], with the blue area being the non-melting region, the area between the two dashed lines being the incongruent melting region, and the orange area being the melting region (excluding the blue line of ZIF-8-mim$_x$im$_y$). **b** Visualization of the glass transition temperatures of the glass forming ZIF-8-mim$_x$im$_y$bim$_z$ derivatives in the ternary phase diagram.

The ZIF-8 derivatives that are close to but not in the compositional region required for congruent melting display incongruent melting, thus leading to crystal-glass-composites (CGCs) after cooling the solid-liquid mixture to room temperature (Supplementary Table 4). For example, ZIF-8-mim$_{0.18}$im$_{0.81}$bim$_{0.01}$ forms a CGC of crystalline ZIF-61 and glassy a$_g$ZIF-8 (Fig. 1b, Supplementary Figs. 45 and 74b) due to the deficiency of bim⁻, while ZIF-8-mim$_{0.43}$im$_{0.38}$bim$_{0.19}$ forms a CGC of crystalline ZIF-8 and glassy a$_g$ZIF-8 (Fig. 1b, Supplementary Figs. 46 and 67b) due to the deficiency of im⁻. Moreover, ZIF-8-mim$_{0.17}$im$_{0.34}$bim$_{0.49}$ maintains the crystalline ZIF-8 phase during thermal treatment, although most of its linkers are exchanged (Supplementary Fig. 48). We conclude that the fraction of im⁻ in this material is too small to cause framework collapse/melting, while the large fraction of bim⁻ supports the crystalline framework.

To investigate if the thermal treatment leads to any chemical alterations of the organic linkers, ¹H NMR (Supplementary Fig. 16) and FTIR spectra were recorded for representative heat-treated ZIF-8 derivatives (Supplementary Fig. 61). Both types of spectroscopic data establish the integrity of the organic linkers after heat treatment and the absence of any organic decomposition products.

The most notable of the above samples is ZIF-8-mim$_{0.15}$im$_{0.74}$bim$_{0.11}$, which features a low $T_m$ of 368 °C in the first DSC upscan, a $T_g$ of 336 °C in the second DSC upscan and the highest $T_g/T_m$ ratio[33,34] of 0.95 (calculated with absolute temperatures) ever reported[35] (Fig. 2a, Supplementary Fig. 74b). The melting point of the linker-exchanged ZIF-8 derivative is drastically lower than the previously predicted "virtual" melting point of 1377 °C for conventional ZIF-8[10]. The enthalpy of melting ($\Delta H_m$) determined by integration of the DSC melting peak is only about 0.7 kJ/mol and thus approximately seven times lower than $\Delta H_m$ of the prototypical ZIF-62[17]. A comparison with the melting enthalpies and entropies of other reported meltable ZIFs can be found in Supplementary Table 3. The very low $\Delta H_m$ of ZIF-8-mim$_{0.15}$im$_{0.74}$bim$_{0.11}$ can be ascribed to its higher porosity, which allows for a more substantial densification during the transition from the crystalline to the liquid state as will be shown by gas sorption data below. Due to increased dispersion interactions in the denser liquid phase, the enthalpy difference between the crystal and the liquid is small. An additional DSC experiment with five heating/cooling cycles in the range from 100 to 430 °C was performed to verify the stability of the glass of ZIF-8-mim$_{0.15}$im$_{0.74}$bim$_{0.11}$ (Supplementary Fig. 76). The fact that $T_g$ only varies between 336 °C and 338 °C in the consecutive upscans implies that a$_g$ZIF-8-mim$_{0.15}$im$_{0.74}$bim$_{0.11}$ has excellent robustness during temperature cycling. Heat capacity ($C_p$) measurements demonstrate that the heat capacity change around the glass

transition ($\Delta C_p$) of a$_g$ZIF-8-mim$_{0.15}$im$_{0.74}$bim$_{0.11}$ is 0.12 J g⁻¹ K⁻¹ (Supplementary Fig. 82); a value comparable to that of a$_g$(IL@ZIF-8) ($\Delta C_p$ = 0.11 J g⁻¹ K⁻¹)[28] and other ZIF glasses (Supplementary Table 5)[10,35]. Moreover, the fragility index ($m$) of a$_g$ZIF-8-mim$_{0.15}$im$_{0.74}$bim$_{0.11}$ obtained from DSC is 21 (Supplementary Fig. 83), a value almost identical to that of fused silica ($m = 20$)[14,35,36], and smaller compared to that of a$_g$ZIF-62 ($m = 23$)[35]. The low fragility is characteristic of a strong liquid that vitrifies to a brittle glass and is in agreement with the extraordinarily high glass-forming ability ($T_g/T_m = 0.95$) of the material.

To exhibit that the SALE approach is not restricted to the im⁻/bim⁻ linker system, we employed the same linker exchange protocol but substituted bim⁻ with Clbim⁻ (5-chlorobenzimidazolate). The derived ZIF-8-mim$_{0.20}$im$_{0.70}$Clbim$_{0.10}$ is again a phase pure ZIF-8 derivative (Supplementary Fig. 36) and can also be transformed into a glass exhibiting a $T_g$ of 322 °C by thermal treatment (Supplementary Figs. 80, 81). Furthermore, with the aim to demonstrate that the SALE approach can also be used to make non-meltable cobalt-based ZIFs meltable, we applied the same linker exchange protocol to ZIF-67, Co(mim)$_2$, which is isostructural to ZIF-8. The derived ZIF-67-mim$_{0.18}$im$_{0.68}$bim$_{0.14}$ is also a phase-pure crystalline framework with **sod** topology, melts at 399 °C and forms a glass with a $T_g$ of 344 °C upon melt-quenching ($T_g/T_m = 0.92$; Supplementary Figs. 50, 55 and 78). This finding suggests that the SALE approach may be a universal tool for the preparation of glasses based on other highly porous ZIFs with other topologies.

## Constructing a triangular phase diagram

Based on the thermal analysis and XRPD data of 50 ZIF-8-mim$_x$im$_y$bim$_z$ derivatives, a ternary phase diagram to visualize the melting and glass-forming properties is constructed (Fig. 3a). The dark orange area highlights the compositional range showing congruent melting and glass formation upon cooling the melt. The blue area marks the compositional range of ZIF-8-mim$_x$im$_y$bim$_z$ derivatives, which do not melt. In the region between the two dashed lines, we observe incongruent melting, leading to a heterogeneous mixture of a liquid ZIF with either a crystalline ZIF with **zni** topology (if the fraction of im⁻ is large) or a crystalline ZIF with **sod** topology (if the fraction of im⁻ is rather small). When the liquid-solid mixtures from the region of incongruent melting are quenched to room temperature, CGCs are formed. This introduces an alternative path for the preparation of CGCs as previously CGCs were formed either from heterogeneous mixtures of two or more MOFs containing a meltable and a non-meltable MOF[37,38] or via physically mixing a crystalline MOF with a prefabricated MOF-glass of

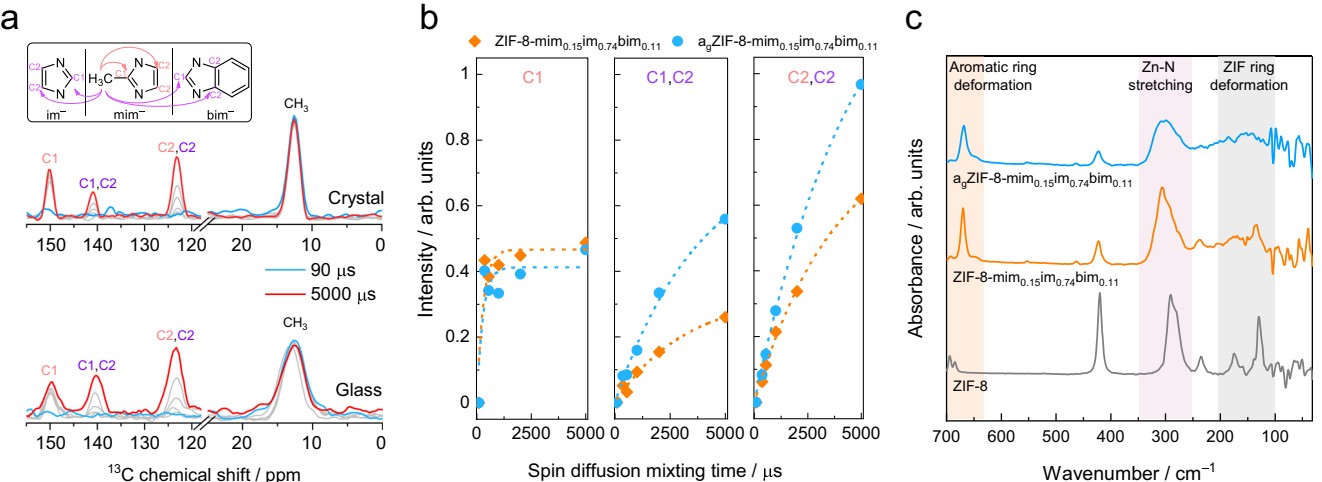

**Fig. 4 | MAS NMR and far-IR spectroscopic characterization. a** Slices along the indirectly detected dimension through the 2D $^1$H-$^{13}$C spin-diffusion solid state NMR spectra of crystalline ZIF-8-mim$_{0.15}$im$_{0.74}$bim$_{0.11}$ (top) and glassy a$_g$ZIF-8-mim$_{0.15}$im$_{0.74}$bim$_{0.11}$ (bottom). Slices were taken at a $^1$H chemical shift of 1.5 ppm and normalized to the peak area of −CH$_3$. The slices show the polarization transfer from the protons of the −CH$_3$ groups of mim$^-$ to all $^{13}$C atoms in the samples as a function of the mixing time. Blue lines represent the slices obtained with a spin-diffusion mixing time of 90 μs, red lines correspond to a mixing time of 5000 μs,

and gradually increasing grey lines correspond to mixing times of 360, 540, 1000 and 2000 μs. **b** Comparison of the spin-diffusion polarization transfer processes in the crystalline and glassy states. Peach-colored labels in a and b represent intra-molecular polarization transfers within mim$^-$, while purple labels represent inter-linker polarization transfers from mim$^-$ to im$^-$ or bim$^-$. Dashed lines are exponential fits to the data. **c** Far-IR spectra of ZIF-8, ZIF-8-mim$_{0.15}$im$_{0.74}$bim$_{0.11}$ and a$_g$ZIF-8-mim$_{0.15}$im$_{0.74}$bim$_{0.11}$.

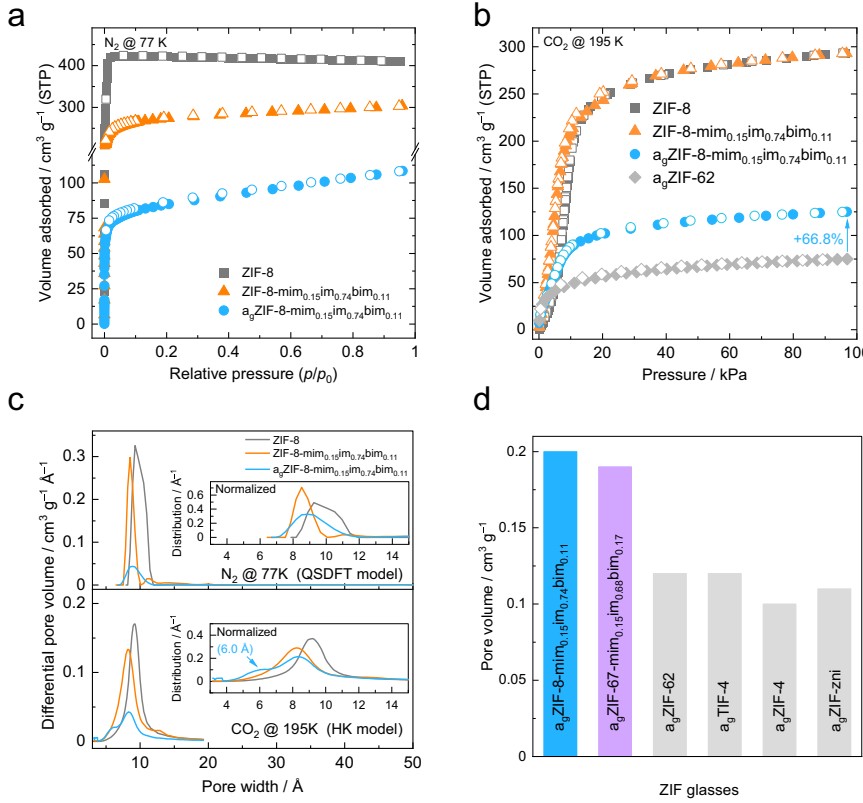

**Fig. 5 | N$_2$ and CO$_2$ gas sorption analyses. a** N$_2$ sorption isotherm at 77 K and **b** CO$_2$ sorption isotherm at 195 K of ZIF-8, ZIF-8-mim$_{0.15}$im$_{0.74}$bim$_{0.11}$ and a$_g$ZIF-8-mim$_{0.15}$im$_{0.74}$bim$_{0.11}$. The CO$_2$ sorption isotherm of a$_g$ZIF-62 recorded at 195 K (taken from ref. 17) is shown for comparison. The solid and hollow circles represent adsorption and desorption, respectively. **c** Pore size distributions for ZIF-8, ZIF-8-mim$_{0.15}$im$_{0.74}$bim$_{0.11}$ and a$_g$ZIF-8-mim$_{0.15}$im$_{0.74}$bim$_{0.11}$ calculated from the N$_2$ adsorption data (QSDFT model, carbon equilibrium transition kernel at

77 K based on a slit-pore model) and the CO$_2$ adsorption data (HK model, iso-therms recorded at 195 K, $p_0$ = 191 kPa, slit-like pore). The insets show normalized pore size distributions for easier visual comparison of the pore widths. **d** Bar plot representing the specific pore volumes calculated from the CO$_2$ sorption capacity at 195 K and 95 kPa for a$_g$ZIF-8-mim$_{0.15}$im$_{0.74}$bim$_{0.11}$, the Co-based a$_g$ZIF-67-mim$_{0.18}$im$_{0.68}$bim$_{0.14}$ and prototypical literature-known ZIF glasses (data taken from ref. 17).

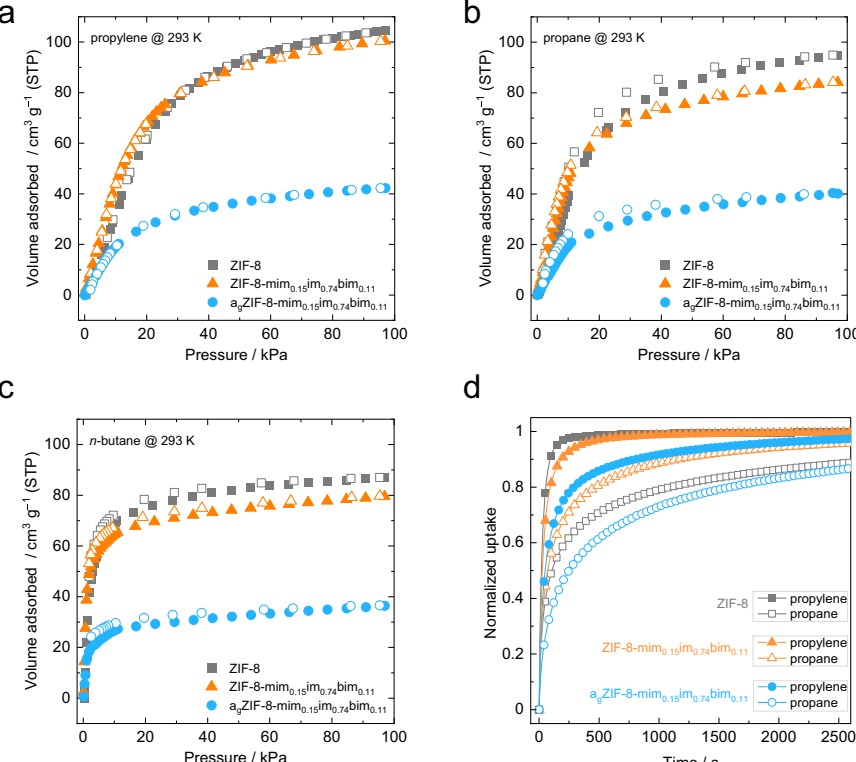

**Fig. 6 | Hydrocarbon sorption studies. a** Propylene sorption isotherms collected at 293 K. **b** Propane sorption isotherms collected at 293 K. **c** *n*-Butane sorption isotherms collected at 293 K. In all panels, adsorption and desorption branches are shown as closed and open symbols, respectively. **d** Kinetic adsorption data of propylene and propane for ZIF-8, ZIF-8-mim$_{0.15}$im$_{0.74}$bim$_{0.11}$ (crystal) and a$_g$ZIF-8-mim$_{0.15}$im$_{0.74}$bim$_{0.11}$ (glass) recorded at 293 K with an equilibrium pressure of about 80 kPa.

the same composition[39]. By contrast, the CGCs are formed from a single phase of mixed-linker ZIF-8 derivatives via thermally triggered phase segregation here.

Besides the compositional influence on the melting behaviour, the molar fractions of the linkers also have a considerable influence on the $T_g$ of the corresponding glass or CGC (Fig. 3b, details in Supplementary Table 4). Depending on *x*, *y* and *z*, $T_g$ ranges between 328 °C and 383 °C. Generally, $T_g$ is lower when the fractions of mim$^-$ and bim$^-$ are low.

## NMR and Far-IR spectroscopy

With the aim of obtaining further insights into the distribution of the three different linkers within the crystal and glass phases of the linker-exchanged ZIF-8 derivatives, 2D $^1$H-$^{13}$C spin-diffusion magic angle spinning (MAS) NMR spectroscopy measurements were recorded on crystalline ZIF-8-mim$_{0.15}$im$_{0.74}$bim$_{0.11}$ and glassy a$_g$ZIF-8-mim$_{0.15}$im$_{0.74}$bim$_{0.11}$ with mixing times between 90 and 5000 μs to track the polarization transfer from the methyl protons of mim$^-$ (resonance at 1.5 ppm) to the various $^{13}$C atoms in the sample (Fig. 4a, Supplementary Figs. 58, 59). Variations in the rate of polarization transfer provide information about the proximity of different organic linkers in the solid[38,40]. With increasing mixing times, polarization transfer from the $-$CH$_3$ protons to the $^{13}$C atoms **C1** and **C2** of the three linkers is observable. The resonance of the **C1** atom of mim$^-$ is isolated at 150 ppm, while **C1** of im$^-$ overlaps with **C1** and **C2** of bim$^-$ (~141 ppm) and **C2** of mim$^-$ and im$^-$ overlap with each other at ~123 ppm. The peak areas of the $-$CH$_3$ resonances were normalized to 1 for both samples to facilitate a comparison of the relative polarization transfer rates to the aromatic carbons. The differences in the rates of fast intra-linker transfers of mim$^-$ (from $-$CH$_3$ to **C1** of mim$^-$) are insignificant when comparing crystal and glass samples

(Fig. 4b, left). In contrast, the inter-linker polarization transfer rate (from $-$CH$_3$ to **C1/C2** of im$^-$ and bim$^-$) in the glass (second panel) is much faster than in the crystalline state. Consistently, the peak representing mixed intra- and inter-linker transfer (third panel) shows a behaviour in between these two cases. This implies that the collapse of the framework structure upon melting, followed by vitrification, leads to a closer intermolecular proximity between the three linkers on average. Although the polarization transfer in the crystalline material is slower compared to the corresponding glass, its high transfer efficiencies suggest that the SALE procedure leads to a rather homogeneous distribution of im$^-$ and bim$^-$ throughout the bulk microcrystals even before melting and not just a clustering of these linkers on the crystal surface.

Far-IR spectroscopy data show that with the partial exchange of mim$^-$ against im$^-$ and bim$^-$, the vibrational bands associated to mim$^-$ (420 cm$^{-1}$ and 677$-$700 cm$^{-1}$) reduce in intensity, while new bands associated to im$^-$ and bim$^-$ (636-681 cm$^{-1}$) emerge in crystalline ZIF-8-mim$_{0.15}$im$_{0.74}$bim$_{0.11}$ (Fig. 4c). Moreover, the band for the stretching vibration of the ZnN$_4$ tetrahedron centred at 291 cm$^{-1}$ in ZIF-8 becomes broader and shifts to 305 cm$^{-1}$ upon linker exchange. This is in agreement with results from a previous far-IR spectroscopy study on crystalline ZIF-4, ZIF-7 and ZIF-8[41], and further supports our reasoning that the im$^-$ and bim$^-$ linkers are rather homogeneously distributed in ZIF-8-mim$_{0.15}$im$_{0.74}$bim$_{0.11}$. For the melt-quenched a$_g$ZIF-8-mim$_{0.15}$im$_{0.74}$bim$_{0.11}$ the stretching band of the ZnN$_4$ tetrahedron is even broader and the characteristic low-energy phonon modes in the region from 100-200 cm$^{-1}$, which are associated with collective distortions of the four- and six-membered rings of the **sod** structure, vanish to a broad continuum[41]. Both observations establish the non-crystalline and highly disordered structure of a$_g$ZIF-8-mim$_{0.15}$im$_{0.74}$bim$_{0.11}$.

## Gas physisorption

Isothermal $N_2$ physisorption at 77 K is an established method for the quantification of the specific pore volume and the specific surface area of crystalline MOFs. All the reported melt-quenched ZIF glasses, however, feature very narrow micropores, which are inaccessible for $N_2$ at 77 K[17]. In contrast to this, $a_g$ZIF-8·$mim_{0.15}im_{0.74}bim_{0.11}$ adsorbs comparatively large amounts of $N_2$ (~108 $cm^3\,g^{-1}$ (STP)) at $p/p_0 = 0.95$) at 77 K and displays a Type I isotherm typical for microporous materials (Fig. 5a, Supplementary Table 6). Although the $N_2$ adsorption capacity near saturation is substantially lower for this ZIF-8 glass compared to ZIF-8 (~411 $cm^3\,g^{-1}$ (STP)) and its crystalline glass-former counterpart ZIF-8·$mim_{0.15}im_{0.74}bim_{0.11}$ (~304 $cm^3\,g^{-1}$ (STP)), to the best of our knowledge, this is the first instance of a ZIF glass adsorbing $N_2$ at 77 K. Pore size distributions (PSDs) calculated from the $N_2$ sorption data (quenched-solid density functional theory, QSDFT model[42]) not only confirm that ZIF-8·$mim_{0.15}im_{0.74}bim_{0.11}$ has a narrower pore size distribution than ZIF-8, but also indicate that a small number of likely disordered sodalite cages still exist in $a_g$ZIF-8·$mim_{0.15}im_{0.74}bim_{0.11}$ (Fig. 5c).

Additional $CO_2$ (kinetic diameter 3.3 Å) sorption measurements recorded at 195 K were performed to get information about the smaller micropores in the materials, which are inaccessible for $N_2$ at 77 K (kinetic diameter 3.6 Å), and so to quantify the full micropore volume ($V_{pore}$)[17]. ZIF-8 and ZIF-8·$mim_{0.15}im_{0.74}bim_{0.11}$ exhibit the same $CO_2$ adsorption capacity of ~293 $cm^3\,g^{-1}$ (STP) at 100 kPa (Fig. 5b and Supplementary Table 6). Naturally, $a_g$ZIF-8·$mim_{0.15}im_{0.74}bim_{0.11}$ adsorbs lower amounts of $CO_2$, however, the capacity at 100 kPa amounts to ~125 $cm^3\,g^{-1}$ (STP), which is about 67% higher than the $CO_2$ capacity of the prototypical glass $a_g$ZIF-62 under identical conditions (Fig. 5b). The BET area ($S_{BET}$) calculated from the $CO_2$ sorption isotherm is 403 $m^2\,g^{-1}$, about twice as large as the BET area of $a_g$ZIF-62 ($S_{BET} = 200\,m^2\,g^{-1}$). PSDs calculated from the $CO_2$ isotherms via the Horvath–Kawazoe (HK) model[43] underline that $a_g$ZIF-8·$mim_{0.15}im_{0.74}bim_{0.11}$ also features a considerable amount of smaller micropores with a diameter of 5–6 Å, which are not present in the crystalline ZIF-8·$mim_{0.15}im_{0.74}bim_{0.11}$ (Fig. 5c). These smaller pores likely result from the collapse of the framework upon melting and glass formation. The specific pore volume of $a_g$ZIF-8·$mim_{0.15}im_{0.74}bim_{0.11}$ calculated from the $CO_2$ sorption data is 0.20 $cm^3\,g^{-1}$, which is by far the highest among ZIF glasses (Fig. 5d). The higher porosity and thus lower density of $a_g$ZIF-8·$mim_{0.15}im_{0.74}bim_{0.11}$ compared to the prototypical $a_g$ZIF-62 is also confirmed by the position of the first scattering peak at smaller $Q$ in the X-ray total scattering data (Supplementary Fig. 57)[44]. Noteworthy, supporting $CO_2$ sorption isotherms recorded on the cobalt-based $a_g$ZIF-67·$mim_{0.18}$-$im_{0.68}$-$bim_{0.14}$ highlight a similarly high pore volume of 0.19 $cm^3\,g^{-1}$ for this glass (Supplementary Figs. 98, 99 and Table 6).

The largely improved porosity of $a_g$ZIF-8·$mim_{0.15}im_{0.74}bim_{0.11}$ compared to previously reported melt-quenched ZIF glasses is further evident by propylene, propane and *n*-butane sorption experiments (Fig. 6 and Supplementary Table 7). Even though the glass adsorbs significantly smaller amounts of these gases than their crystalline relatives, the hydrocarbon sorption capacities are about 3 to 5 times higher compared to other ZIF glasses[17]. Moreover, the hysteresis between adsorption and desorption is small, suggesting negligible diffusion limitations for the larger hydrocarbon molecules.

Kinetic adsorption data showcase that propylene is adsorbed much faster than propane in $a_g$ZIF-8·$mim_{0.15}im_{0.74}bim_{0.11}$, similar to the crystalline ZIF-8 and ZIF-8·$mim_{0.15}im_{0.74}bim_{0.11}$ (Fig. 6d). Fitting the kinetic data using stretched exponential models (Supplementary Figs. 102–110) yielded the corresponding rate constants ($k$, Supplementary Table 9). For crystalline ZIF-8, $k_{propylene}$ and $k_{propane}$ are $3.26 \times 10^{-2}\,s^{-1}$ and $3.18 \times 10^{-3}\,s^{-1}$, resulting in a high kinetic propylene/propane selectivity ($S_{kin} = k_{propylene}/k_{propane}$) of 10.3, as expected from previous reports[45]. By contrast, $k_{propylene}$ and $k_{propane}$ are more similar for ZIF-8·$mim_{0.15}im_{0.74}bim_{0.11}$ ($2.14 \times 10^{-2}$ and $6.21 \times 10^{-3}\,s^{-1}$),

resulting in a much lower $S_{kin}$ of just 3.5. Hence, the structural complexity of the pores caused by the three imidazolate-type linkers present after SALE seems to reduce the selectivity. Strikingly, the corresponding glass $a_g$ZIF-8·$mim_{0.15}im_{0.74}bim_{0.11}$ features a significantly improved selectivity ($S_{kin} = 5.4$) even though the rate constants are slightly lower than for its crystalline relative ($k_{propylene} = 9.37 \times 10^{-3}\,s^{-1}$ and $k_{propane} = 1.73 \times 10^{-3}\,s^{-1}$) as expected due to the lower porosity and narrower pore sizes. These findings highlight that glass formation can be a beneficial tool not only for materials processing but also for improving the functional sorption characteristics of a porous material.

## Discussion

This work demonstrates that solvent-assisted linker exchange is an effective strategy to make non-meltable, highly porous MOFs meltable. In the case of ZIF-8, exchange of the strongly coordinating $mim^-$ linkers by two different linkers is necessary to obtain meltable and glass-forming derivatives with an isoreticular structure of **sod** topology. Exchange of $mim^-$ for the smaller and weaker coordinating $im^-$ linker facilitates metal-linker bond breaking, while the parallel exchange for the larger $bim^-$ linker stabilizes the liquid phase and prevents the crystallization of a dense ZIF. Using a systematic approach, we were able to screen the compositional phase space to create a ternary phase diagram visualizing the regions of congruent melting, incongruent melting, or no melting. Some linker-exchanged ZIF-8 derivatives exhibit ultra-high glass-forming ability ($T_g/T_m = 0.95$) and specific pore volumes substantially larger than those of other MOF glasses. The high porosity of the ZIF-8 glass is visualized not only by the adsorption of $N_2$ at cryogenic temperatures but also by high sorption capacities for hydrocarbon gases and very good selectivities for the kinetic separation of propylene from propane. These significantly improved properties of ZIF-8 glasses compared to other MOF glasses make them attractive materials for applications in high-performance gas separation membranes, where the grain-boundary-free structure of MOF glasses is particularly advantageous[46,47]. Given that the number of meltable MOFs is still very limited, the linker exchange strategy presented here reveals important prospects for the further development of meltable and glass-forming MOFs based on the very large number of literature-known non-meltable frameworks. Applying this strategy to other MOFs with higher porosity than ZIF-8 may lead to even more porous MOF glasses with porosity features comparable to their crystalline congeners.

## Methods

### Materials synthesis

ZIF-8 was synthesized using the method previously reported in the literature with minor modifications[21,37]. Zinc nitrate hexahydrate (2.1 g, 7.06 mmol) was added to 2-methylimidazole (1.2 g, 11.62 mmol) and both powders were dissolved in *N,N*-dimethylformamide (DMF) (80 mL). The mixture was covered and stirred for 30 min at room temperature to ensure a homogenous solution and then heated at 120 °C for 24 h. The obtained precipitate was collected by filtration and soaked in dichloromethane (DCM) (80 mL) for 24 h. Subsequently, dynamic vacuum activation ($p \approx 10^{-4}$ kPa) was used at 170 °C for 6 h to obtain activated ZIF-8. ZIF-67 was synthesized using the same method but replacing zinc nitrate hexahydrate with cobalt nitrate hexahydrate.

Synthesis of ZIF-8·$mim_x im_y bim_z$ is based on the exchange method of SALEM-2[21]. Imidazole, benzimidazole, or a mixture of imidazole and benzimidazole (see Supplementary Information Section 1 for the corresponding specific amount) was placed in a 20 mL microwave vial and dissolved in *n*-butanol (20 mL) by sonication. Activated ZIF-8 crystals (100 mg, 0.44 mmol) were immersed in the resulting solution. The vial was capped and placed in an isothermal oven at 100 °C for 1, 3, 5 and 7 days. After decanting off the solvent and washing with 5 mL of DCM,

the samples were transferred to a dynamic vacuum ($p \approx 10^{-4}$ kPa) for activation at 100 °C for 12 h.

Based on the above crystalline materials, their thermal products were obtained via thermal treatment in a TGA/DSC equipment (see Supplementary Information Section 6 for further details).

### X-ray powder diffraction (XRPD)
XRPD patterns were recorded at room temperature on a Siemens D5005 diffractometer or a Bruker D8 Advance diffractometer. Data were collected with CuKα radiation in the range from 5° to 50° 2θ with a step size of 0.02°. Finely ground samples (crystalline or glassy) were deposited on a glass holder or a single crystal zero background sample holder made of silicon (cut along the (610) plane). For phase identification, structureless profile fits (Pawley method[30]) were performed with the TOPAS academic v6 software[48].

### Variable temperature XRPD (VT-XRPD)
X-ray powder diffraction data at various temperatures were collected at BL9 of DELTA (Dortmunder Elektronenspeicherring-Anlage, Dortmund, Germany) with a monochromatic X-ray beam (λ = 0.45920 Å) using a MAR345 image plate detector. Finely ground samples were sealed in quartz capillaries (diameter 1 mm) in an $N_2$-filled glovebox and heated using an Anton Parr heating stage covered with a graphite dome from 27 °C to a temperature before $T_d$ (previously determined via TGA). Temperature calibration of the heating stage was performed by reference XRPD measurements of α-quartz. Data were integrated using the DAWN[49] software package.

### FTIR spectra spectroscopy
Fourier transform infrared (FTIR) spectroscopies of both MIR (mid IR, $\tilde{\nu} = 4000\ cm^{-1}$–$400\ cm^{-1}$) and FIR (far IR, $\tilde{\nu} = 700\ cm^{-1}$–$30\ cm^{-1}$) were performed on a Spectrum 3 FTIR spectrometer from Perkin Elmer equipped with a Gladi ATR-300 unit from Pike Technologies (ATR = attenuated total reflectance). After scanning the background, powdered samples were placed on the diamond ATR unit and carefully compressed with a stamp for the measurement.

### Nuclear magnetic resonance (NMR) spectroscopy
Solid-state NMR experiments were carried out on a 700 MHz NMR spectrometer system equipped with a 1.3 mm probe. For two-dimensional $^1H$–$^{13}C$ spin-diffusion measurements, the number of scans was 56, and the number of increments in indirectly detected dimensions was 888, with a spectral width of 27.777 kHz for $^1H$ dimension, and 55.555 kHz for $^{13}C$ dimension. Repetition delay between scans was 0.8 s. The data were processed with the TopSpin (v4.1.4) software. Solution $^1H$ NMR spectroscopy was performed on digested crystalline and glassy ZIF samples with Bruker DPX-300, DPX 500 or Agilent DD2 500 spectrometers. The solid samples were digested before the measurement using DMSO-$d_6$ (0.5 mL) and DCl/$D_2O$ (35 wt%, 0.015 mL) as solvents. The data were processed with the MestReNova (v14.2.0) software. Data were referenced to the residual proton signal of DMSO.

### Thermal analysis
Differential scanning calorimetry (DSC) measurements were performed on a DSC 25 from TA Instruments under a constant nitrogen flow (50 mL min$^{-1}$). For all measurements, a heating/cooling rate of ±10 °C min$^{-1}$ was applied (except for the variable heating/cooling rate measurements for the determination of the fragility and the determination of heat capacity by modulated DSC; see Supplementary Information). Before the measurement, the samples were ground and placed in a hermetically sealed aluminium crucible, and a hole was pinched into the lid of the sealed crucible. Thermogravimetric analysis (TGA) measurements were conducted on an SDT 650 from TA

Instruments under a constant nitrogen flow (100 mL min$^{-1}$). Ceramic alumina crucibles (90 μL) were used for the TGA measurements. The heating rate was 10 °C min$^{-1}$. All thermal analysis data were processed and evaluated using the TRIOS (v5.1.0.46403) software from TA instruments. According to previous studies, the melting point ($T_m$) is defined as the offset temperature of the calorimetric melting peak in the first DSC upscan and the glass transition temperature ($T_g$) is specified as the onset point of the endothermic step in the second DSC upscan, whereas all other derived temperatures are defined as the peak temperature[10,11,14,35].

### Isothermal gas physisorption
Experiments were performed with a Quantachrome iQ MP porosimeter. Sample quantities of about 100 mg (for crystals) and at least 40 mg (for all others) were used for the experiments. Prior to the first measurement, the ground samples were degassed under a dynamic vacuum ($p \approx 10^{-5}$ kPa) at 100 °C for 2 h. Gas sorption isotherms were measured with $N_2$ (77 K, gas purity > 99.999%), $CO_2$ (195 K, dry ice/isopropanol bath, gas purity > 99.995%) and n-butane (273 K and 293 K, gas purity > 99.95%), propane and propylene (293 K, gas purity > 99.95%). Between measurements, samples were degassed under a dynamic vacuum ($p \approx 10^{-5}$ kPa) at ambient temperature for approximately 2 h. After adsorption measurements with n-butane, the samples were again heated to 100 °C for 30 min under a dynamic vacuum ($p \approx 10^{-5}$ kPa). Pore size distributions were determined from the $N_2$ adsorption data (QSDFT model, carbon equilibrium transition kernel at 77 K based on a slit-pores model) and the $CO_2$ adsorption data (HK model, isotherms recorded at 195 K, $p_0 = 191$ kPa, slit-like pore model) using the ASiQwin software package. Kinetic hydrocarbon gas adsorption data were also recorded with a Quantachrome iQ MP porosimeter at 293 K with an equilibrium pressure of about 80 kPa.

### X-ray total scattering
X-ray total scattering data were collected at beamline P02.1 at Deutsches Elektronen-Synchrotron (DESY, Germany) using a monochromatic X-ray beam (λ = 0.20734 Å, 60 keV) and a Perkin Elmer XRD1621 (2048 × 2048 pixels active area) detector. The finely ground samples were placed in 1 mm (outer diameter) quartz glass capillaries. For the VT experiments, a hot air blower was used to heat the samples from room temperature to a maximum of 530 °C. The heating rate between the various set temperatures was 20 °C min$^{-1}$. After reaching the set temperatures, 10 consecutive diffraction patterns were collected with continuous rotation of the sample and an exposure time of 60 s. These 10 diffraction patterns were averaged for data analysis. For all datasets, background subtraction was performed with scattering data collected from an empty capillary. Background subtraction and corrections for multiple, container and Compton scattering, as well as for absorption, were done with the GudrunX program. The normalized scattering functions ($S(Q)$) were Fourier transformed to yield the pair distribution functions (PDFs) in the form of $D(r)$[31,32].

### Scanning electron microscopy (SEM)
SEM imaging was performed with a Hitachi S-4500 instrument. For measurements, samples were placed on a conductive adhesive pad. Imaging was done with 1 kV accelerating voltage on a secondary electron detector. All investigated samples were ground and taken from the sorption tubes after conduction of physisorption measurements before imaging.

## Data availability
The authors declare that all data supporting the findings of this study are available within the article and its supplementary information files. The corresponding raw data are available on request from the corresponding author S.H.

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

## Acknowledgements

This project received funding from Deutsche Forschungsgemeinschaft within the project 447344931 (HE 7628/7-1). W.-L.X acknowledges the China Scholarship Council (CSC No. 202008110211). P.K. acknowledges the Fonds der Chemischen Industrie for a Kekulé-Fellowship. The authors thank DELTA Dortmund for allocation of beamtime at beamline BL9 and Dr. Christian Sternemann and Dr. Michael Paulus for their help with the variable temperature XRPD experiments. We acknowledge the DESY (Hamburg, Germany), a member of the Helmholtz Association HGF, for the provision of experimental facilities. X-ray total scattering data were carried out at PETRA III on beamline P02.1 (proposal I-20210316 and I-20220350). Volker Brand is acknowledged for the operation of the SEM instrument.

## Author contributions

S.H. and W.-L.X. designed the project. W.-L.X. synthesized the materials and performed and analyzed XRPD, FTIR spectroscopy, $^1$H NMR spectroscopy and thermal analysis and gas sorption experiments. J.S. contributed to the preparation of ZIF-8. W.-L.X. performed the synchrotron XRPD experiments with the help of P.K., A.K., R.P. and L.F-B. P.K., A.K., and R.P. collected X-ray total scattering data. W.-L.X. processed and analyzed all the synchrotron radiation data with the help of J.S. H.A. and S.V. performed the solid-state NMR experiments with R.L. contributing to data analysis and interpretation. W.-L.X and S.H. wrote the manuscript with contributions from all the authors. All authors have given approval of the final manuscript.

## Funding

## Competing interests

The authors declare no competing interests.
