## [Peer Review File · Nature Communications]

Highly porous metal-organic framework liquids and glasses via a solvent-assisted linker exchange strategy of ZIF-8This manuscript has been previously reviewed at another journal that is not operating a transparent peer review scheme. This document only contains reviewer comments and rebuttal letters for versions considered at *Nature Communications*.

REVIEWER COMMENTS

Reviewer #1 (Remarks to the Author):

The authors have modified their manuscript to address many of my previous questions and concerns. This manuscript is likely suitable for publication in Nature Communications, but it would be ideal if the authors could more adequately address the following points:

1) It is still not particularly convincing that ZIF-8-mim_{0.15}m_{0.74}bim_{0.11} undergoes a "melting" transition (as opposed to amorphization). This is important as the authors underscore that SALE technique enabled the melting transition of pristine sodalite ZIF structure for the first time. The extremely small ΔH_{fus} and ΔS_{fus} (0.7 kJ/mol and 1.03 J/K•mol, respectively) is particularly concerning. In Supplementary Figure 64 (DSC of ZIF-8-mim_{0.18}m_{0.72}bim_{0.10}) and Supplementary Figure 80 (DSC of ZIF-8-mim_{0.20}m_{0.70}cbim_{0.10}), there is an exothermic feature before the endothermic feature assigned to melting. Based on the authors' explanation and the DSC results, the exothermic framework collapse and the endothermic melting transition seem to occur simultaneously for ZIF-8-mim_{0.15}m_{0.74}bim_{0.11}. In this case, the measured ΔH of this transition should include the exothermic ΔH for the framework collapse, thus the actual ΔH_{fus} and ΔS_{fus} of this melting transition should be larger than the measured ΔH , which seems more reasonable.

Additionally, can the authors comment on if the heat capacity changes for ag-ZIF-8-mim_{0.15}m_{0.74}bim_{0.11} around the glass transition temperature are comparable to the reported values for a glass-to-liquid transition (e.g., 0.11 J/g•K for IL@ZIF-8, Nat. Commun. 2021, 12, 5703)?

2) With respect to propylene/propane selectivity, both diffusivity and solubility are important for membrane applications, and it is difficult to determine whether or not this glass might actually be useful for membrane applications. Moreover, mixed-gas diffusivity differences could be different than pure-component ones.

Reviewer #2 (Remarks to the Author):

In reviewing this manuscript again I have calibrated my comments to the obvious fact that the bar for novelty and significance of a paper is much, much lower for Nature Communications than for Nature Materials (the original journal).

Re-reading my own comments and those of the other reviewers I think it was the correct decision to reject the paper from Nature Materials. However, for Nature Communications this is a much closer decision, and indeed I probably come down in favour of acceptance of this particular submission.

The main reason for initial rejection was that the conceptual novelty of the work was compromised by previous publications (some by the same group). All referees pointed this out. For Nature Communications I think the main question becomes whether the increase in porosity (almost two-fold) is significant enough in itself to merit publication. I think that for Nature Communications this is a significant enough result. Increasing the porosity of such glassy materials is clearly of great importance for many (although not all) of their potential applications. The fact that a major jump in porosity is possible by changing composition will be of interest to a wide range of the readership interested in framework materials.

There was never really a question about the technical merit of the paper and the questions that were raised (e.g. the weak DSC signals) have been answered suitably by the authors. Therefore I think that the paper is suitable for publication in Nature Communications.

RESPONSE TO REVIEWERS' COMMENTS

Reviewer #1 (Remarks to the Author):

The authors have modified their manuscript to address many of my previous questions and concerns. This manuscript is likely suitable for publication in Nature Communications, but it would be ideal if the authors could more adequately address the following points:

Response:

We thank the reviewer for the positive assessment of our revised manuscript.

1) It is still not particularly convincing that ZIF-8-mim_{0.15}im_{0.74}bim_{0.11} undergoes a “melting” transition (as opposed to amorphization). This is important as the authors underscore that SALE technique enabled the melting transition of pristine sodalite ZIF structure for the first time. The extremely small ΔH_{fus} and ΔS_{fus} (0.7 kJ/mol and 1.03 J/K•mol, respectively) is particularly concerning. In Supplementary Figure 64 (DSC of ZIF-8-mim_{0.18}im_{0.72}bim_{0.10}) and Supplementary Figure 80 (DSC of ZIF-8-mim_{0.20}im_{0.70}Clbim_{0.10}), there is an exothermic feature before the endothermic feature assigned to melting. Based on the authors' explanation and the DSC results, the exothermic framework collapse and the endothermic melting transition seem to occur simultaneously for ZIF-8-mim_{0.15}im_{0.74}bim_{0.11}. In this case, the measured ΔH of this transition should include the exothermic ΔH for the framework collapse, thus the actual ΔH_{fus} and ΔS_{fus} of this melting transition should be larger than the measured ΔH , which seems more reasonable.

Response:

This is a very interesting point. Given that macroscopic flow is evident for a_gZIF-8-mim_{0.15}im_{0.74}bim_{0.11}, the parent material ZIF-8-mim_{0.15}im_{0.74}bim_{0.11} is clearly in a liquid state when heated across the melting point. Indeed, framework collapse and melting happen simultaneously. Hence, it is not possible to separate the two events. It is reasonable that the framework collapses because of the metal-linker-bond dissociation associated with melting. So, framework collapse and melting are closely intertwined and likely not independent events (at least on the timescale of the current experiments). The same is observed for ZIF-62 and TIF-4, which both undergo densification (i.e. a partial framework collapse) upon melting (see *Nat. Commun.* **2022**, 13, 7750). As outlined in our manuscript, the magnitude of densification at the solid-liquid transition is much more significant for ZIF-8-mim_{0.15}im_{0.74}bim_{0.11} than for ZIF-62 and TIF-4. Thus, it is reasonable that the enthalpy and entropy of fusion of ZIF-8-mim_{0.15}im_{0.74}bim_{0.11} are substantially smaller than the ones of ZIF-62 and TIF-4.

Additionally, can the authors comment on if the heat capacity changes for a_g-ZIF-8-mim_{0.15}im_{0.74}bim_{0.11} around the glass transition temperature are comparable to the reported values for a glass-to-liquid transition (e.g., 0.11 J/g•K for IL@ZIF-8, *Nat. Commun.* 2021, 12, 5703)?

Response:

Thank you for this suggestion. We now determined the heat capacity (C_p) of a_gZIF-8-mim_{0.15}im_{0.74}bim_{0.11} in the temperature range from 200 to 415 °C and we compared the heat capacity change (ΔC_p) of a_gZIF-8-mim_{0.15}im_{0.74}bim_{0.11} around its glass transition (Supplementary Figure 82 in the revised Supplementary Information File) with other reported representative ZIF glasses (Supplementary Table 5 in the revised Supplementary Information File). ΔC_p amounts to 0.12 J g⁻¹ K⁻¹ and is similar to that of a_g(IL@ZIF-8-HT) ($\Delta C_p = 0.11$ J g⁻¹ K⁻¹) and that of other ZIF glasses previously reported. We added a new section with the heat capacity data to the Supplementary Information File (Supplementary Methods 6.2, page 52 in the revised Supplementary Information File) and an additional statement to the revised manuscript.

Addition to the manuscript text (page 10):

Heat capacity (C_p) measurements demonstrate that the heat capacity change around the glass transition (ΔC_p) of $a_g\text{ZIF-8-mim}_{0.15}\text{im}_{0.74}\text{bim}_{0.11}$ is $0.12 \text{ J g}^{-1} \text{ K}^{-1}$ (Supplementary Figure 82); a value comparable to that of $a_g(\text{IL@ZIF-8})$ ($\Delta C_p = 0.11 \text{ J g}^{-1} \text{ K}^{-1}$) and other ZIF glasses (Supplementary Table 5).^{10,36}

Addition to the Supplementary Information File (page 52):

Supplementary Methods 6.2 – Heat capacity measurements

The evolution of heat capacity (C_p) of $a_g\text{ZIF-8-mim}_{0.15}\text{im}_{0.74}\text{bim}_{0.11}$ in the range from 200 to 415 °C was determined by modulated DSC using a DSC 25 calorimeter (TA Instruments). In this measurement, a sinusoidal modulation with a temperature amplitude of $\pm 1 \text{ °C}$ and a modulation period of 120 s was overlaid on a linear heating ramp with an average heating rate of 2 °C min^{-1} . Baseline and sapphire reference scans were collected before the sample scan using the same temperature program.

Supplementary Figure 82. Heat capacity (C_p) scan of $a_g\text{ZIF-8-mim}_{0.15}\text{im}_{0.74}\text{bim}_{0.11}$. The heat capacity change around the glass transition (ΔC_p) was determined using the difference between the two intersections of the onset and offset tangent lines of the glass transition signal.

Supplementary Table 5. Comparison of the heat capacity change (ΔC_p) around the glass transition of $a_g\text{ZIF-8-mim}_{0.15}\text{im}_{0.74}\text{bim}_{0.11}$ and other reported ZIF glasses.

Material	Composition	T_g (°C)	ΔC_p ($\text{J g}^{-1} \text{ K}^{-1}$)	References
$a_g\text{ZIF-8-mim}_{0.15}\text{im}_{0.74}\text{bim}_{0.11}$	$\text{Zn}(\text{mim})_{0.30}(\text{im})_{1.48}(\text{bim})_{0.22}$	336	0.12	This work
$a_g\text{ZIF-62}$	$\text{Zn}(\text{im})_{1.75}(\text{bim})_{0.25}$	322	0.19	Sci. Adv. 4, eaao6827 (2018)
$a_g(\text{IL@ZIF-8})$	$[\text{EMIM}][\text{TFSI}]\text{@Zn}(\text{mim})_2$	322	0.11	Nat. Commun. 12, 5703 (2021)
$a_g\text{ZIF-4 (HDA)}$	$\text{Zn}(\text{im})_2$	292	0.16	Nat. Commun. 6, 8079 (2015)
$a_g\text{ZIF-4 (LDA)}$		316	0.11	

2) With respect to propylene/propane selectivity, both diffusivity and solubility are important for membrane applications, and it is difficult to determine whether or not this glass might actually be useful for membrane applications. Moreover, mixed-gas diffusivity differences could be different than pure-component ones.

Response:

We agree with the reviewer. Indeed, membrane permeability is determined by both diffusivity and solubility. The solubility (adsorption capacity) of propylene in $a_g\text{ZIF-8-mim}_{0.15}\text{im}_{0.74}\text{bim}_{0.11}$ is only slightly higher than that of propane (similar to crystalline ZIF-8 but with a generally lower capacity), but the sorption kinetics of propylene are significantly faster than the ones of propane. As mentioned in our manuscript, this finding suggests a potential for the kinetic separation of these two gases with an $a_g\text{ZIF-8-mim}_{0.15}\text{im}_{0.74}\text{bim}_{0.11}$ -based membrane. Membrane preparation and gas permeation tests will be performed in a follow-up work.

Reviewer #2 (Remarks to the Author):

In reviewing this manuscript again I have calibrated my comments to the obvious fact that the bar for novelty and significance of a paper is much, much lower for Nature Communications than for Nature Materials (the original journal).

Re-reading my own comments and those of the other reviewers I think it was the correct decision to reject the paper from Nature Materials. However, for Nature Communications this is a much closer decision, and indeed I probably come down in favour of acceptance of this particular submission.

The main reason for initial rejection was that the conceptual novelty of the work was compromised by previous publications (some by the same group). All referees pointed this out. For Nature Communications I think the main question becomes whether the increase in porosity (almost two-fold) is significant enough in itself to merit publication. I think that for Nature Communications this is a significant enough result. Increasing the porosity of such glassy materials is clearly of great importance for many (although not all) of their potential applications. The fact that a major jump in porosity is possible by changing composition will be of interest to a wide range of the readership interested in framework materials.

There was never really a question about the technical merit of the paper and the questions that were raised (e.g. the weak DSC signals) have been answered suitably by the authors. Therefore I think that the paper is suitable for publication in Nature Communications.

Response:

We are grateful to the reviewer for the time they have taken to read the revised manuscript and thank the reviewer for the positive comments.

REVIEWERS' COMMENTS

Reviewer #1 (Remarks to the Author):

The authors have addressed all of my concerns in their revised manuscript.